# Does Prolonged FFP2 Mask Use Cause Changes in Nasal Cytology? A Pilot Observational Study on Healthcare Workers

**DOI:** 10.3390/healthcare10122365

**Published:** 2022-11-24

**Authors:** Massimo Campagnoli, Valeria Dell’Era, Maria Silvia Rosa, Paolo Aluffi Valletti, Massimiliano Garzaro

**Affiliations:** ENT Division, University of Eastern Piedmont, c.so Mazzini 18, 28100 Novara, Italy

**Keywords:** COVID-19, FFP2 mask, adverse effect, face mask, nasal cytology, healthcare workers, nasal symptoms, healthcare professionals, practices

## Abstract

TheCOVID-19 pandemic has rapidly spread worldwide. Individual prevention approaches include FFP2/N95 mask use. Healthcare (HC) workers wear face masks for a long time during their work shifts and often complain of nasal symptoms. Current data on mask-associated symptoms or upper airway epithelium transformations are limited. Nasal cytology (NC) is a useful, non-invasive diagnostic method to assess cellular alterations. The aim of this study is to compare NC in HC workers before and after the continuous wearing of FFP2 face masks. We conducted a pilot observational study on 10 volunteer HC workers, who continuously used FFP2 masks during the work shift. All subjects underwent NC at the beginning (T0) and at the end of their workshift (T1) and the cytological findings were compared. Moreover, nasal symptoms were collected. Rare inflammatory cells were detected at T0 and, comparing cytological data about T0 and T1, no significant differences were observed. The most reported nasal symptoms were itching (70%) and a dry nose (60%). Difficulty of breathing and nasal blockage were not relevant. These preliminary data seem to suggest that wearing an FFP2 mask does not determine observable alterations in NC in daily work. However, further studies on a larger population for a longer period are needed.

## 1. Introduction

The COVID-19 pandemic has deeply changed habits and behavior among healthcare (HC) workers [1,2]. One of the most impactful changes in personal protective equipment is probably the frequent employment of FFP2/N95 filtering masks in order to reduce the risk of contagion from COVID-19 [3]. FFP2 masks are used both by the HC workers and general population and their employment was found to be one of the most effective measures in preventing SARS-CoV-2 infection and limiting its spreading [4,5]. However, some side effects have been reported to be associated with the use of this kind of mask, especially in the dermatological and pneumological field [6,7]. The use of FPP2 masks creates a warm and humid environment because of their barrier effect. In the literature, a significant increase in heat flow and perioral region temperature is reported when wearing these FFP2 respirators [8,9,10]. Moreover, a severe increase in humidity due to the condensation of the exhaled air could determine changes in the natural skin milieu, especially of perioral and perinasal areas [11,12]. HC workers often complain of difficulty in breathing, nasal blockage, rhinorrhea, sneezing, itching and dry nose [13]. Klimek et al. reported statistically significant evidence of mask-induced rhinitis, itching and swelling of the mucous membranes, as well as increased sneezing. Endoscopically, an increased secretion was recorded, suggesting the presence of inhaled mask polypropylene fibers as the trigger of mucosal irritation [14]. Prolonged warmer breathing and humidity may induce transformation (the so-called “adaptation effect”) in the upper airway epithelium and subsequent related symptoms. These cellular alterations could be safely and easily investigated using nasal cytology [15]. Nasal cytology is a useful, cheap, non-invasive and easy-to-apply diagnostic method to assess nasal inflammation and disease-specific cellular features [15]. The aim of the current study is to compare nasal cytology in HC workers before and after continuous wearing of FFP2 face masks for acontinuous 8 h period.

## 2. Materials and Methods

This pilot observational study reported data about 10 volunteer HC workers employed at Novara Maggiore Hospital between May and June 2022.Each subject underwent a nasal scraping with a Rhinoprobe™ at the beginning (T0) and at the end of his work shift (T1) (minimum 8 h shift). The examination was performed bilaterally in order to provide an adequate sample. The Rhinoprobe™ was scraped above the inferior turbinate, collecting mucus full of the exfoliating cells of the nasal epithelium. Microscopic slides were settled by rubbing the probe on them. All specimens were spread on a microscope slide and airdried. May-Grunwald Giemsa Quick stain was employed to stain all samples. All nasal scrapings were performed by the same operator and were analyzed blinded by an otolaringologist expert in nasal cytology. The presence of red blood cells was considered an exclusion criterion and required a new sample: the presence of blood indicates the possibility of a vessel rupture, determining the contamination of the nasal epithelium by inflammatory blood cells. The microscope observation required both qualitative and quantitative examination. The first general evaluation was based on the chromatic features of the sample, e.g., purple predominance suggests the presence of neutrophilic infiltration, whereas red prevalence instead indicates the presence of eosinophil cells. The second step is to compare the dimensions and morphology of the cellular elements of the sample. Concerning these two aspects it is important to observe that ciliated epithelial cells are bigger and taller than inflammatory cells, which appear rounded and smaller in dimension. After this, it is important to observe the location of granules (they can be inside or outside the cells); these elements can be very useful for clinical interpretation. Visualizing diffuse red granules outside eosinophil cells suggests histamine release, which can lead to symptoms such as sneezing and itching. Considering all these elements, the observer will be able to identify the different cellular populations. In this study we focused on the following:-Lymphocytes-Neutrophils-Eosinophils-Mast cells-Hair cells-Muciparous cells

Moreover, it is important to count the number of cells in each high-power field (hpf). Due to the unavailability of a phase-contrast electronic microscope, no motility analysis of ciliated cells was performed. A total of 20 specimens were collected. The comparison between the cytological aspects of T0 and T1 samples was performed using the attached form (Figure 1). According to aforementioned criteria, nasal cytology was employed to identify the normal cells (ciliated and mucinous), the inflammatory cells (lymphocytes, neutrophils, eosinophils, mast cells), bacteria or fungal hyphae/spores. Apart from the normal cell population, some specific cytological patterns can be useful in discriminating among various diseases (15). Each cellular type is quantified and its numerosity describes a severity score (e.g., eosinophils +: 1–5 cells, ++: 6–10 cells, +++: 11–30 cells) [16]. We considered as significant any variation in the number of investigated cells that caused a change in severity score. Moreover, nasal symptoms (difficulty in breathing, dry nose, nasal blockage, rhinorrhea, epistaxis, sneezing and itching) were collected in the cohort using a self-reported symptom questionnaire. Each volunteer could report more than one symptom.

## 3. Results

Clinical and cytological data about the 10 HC workers were collected. Males were prevalent in the group (six males and four females). The mean age of the group was 31.3 ± 5.98 (min 26; max 46). All volunteers were Caucasian; only three were active smokers and no one used nasal spray (either steroids or vasoconstrictor). All volunteers had no history of sino-nasal disease/surgery. The most reported nasal symptoms were itching (70%) and dry nose (60%). The most frequent combination of symptoms was itching, dryness and sneezing.

Difficulty in breathing and nasal blockage were complained of by 20% of the sample. One HC worker did not complain about nasal symptoms after wearing an FPP2 mask.

All referred symptoms are collected in Figure 2. Concerning cytological findings (Figure 3), no bacteria were observed in any of the specimens; rare inflammatory cells were detected at T0. Such findings did not vary at T1 control. In particular, the presence of eosinophils was detected in the specimens of only three volunteers (two low-grade [+] and one moderate-grade [++]), but the count of these inflammatory cells did not change between T0 and T1.A low-grade (+) mast cell infiltration was observed only in one HC worker with no difference between the two observations. As far as the neutrophil count is concerned, two cases presented a moderate infiltration (++) and one low infiltration (+), and this did not differ during the time. In all specimens, ciliated cells did not show morphological alterations, suggesting viral infection such as loss of cilia, chromatin thickening and cytoplasm, including body. Ciliated cells and muciparous cells showed a mild decrease in the absolute count after the wearing of an FFP2 mask for an 8-h work shift; however, this variation was not sufficient to determine a change in the severity score of the cell count.

All detailed cytological features are reported in Table 1.

## 4. Discussion

Wearing FFP2 masks has been demonstrated to be one of the most effective precautions in order to avoid SARS-CoV2 infection. Moreover, it has to be considered that wearing an FFP2 mask has become essential, especially for HC workers performing aerosol-generating procedures. Prolonged use of FFP2 masks has been reported as a risk factor for developing mainly dermatological and pneumological complications (e.g., COPD exacerbation, dermatitis) [17,18].In a recent work by Battista et al. conducted on a cohort of 185 HC workers using FFP2 masks for at least 6 consecutive hours, more than 80% of them reported nasal symptoms such as obstruction or dyspnea, dry nose or crusting, sneezing or runny nose and nasal itching [19].The aim of the current research was to evaluate if the continuous use of FFP2 masks could induce alterations in nasal cytology of HC workers in a short observational time period. Comparing the cytological data ofT0 and T1, no significant differences were observed in the examined group. A mild decrease in the number of hairy cells and muciparous cells was noticed without the severity score changing. No increase in inflammatory cell count (eosinophils, mast cells, neutrophils) was observed after 8 h wearing an FFP2 mask. These data seem to suggest that wearing an FFP2 mask does not determine observable alterations in nasal cytology in daily work. In the current study, it was observed that symptoms were present in the great majority of the volunteers; in fact, in only one case were symptoms not reported. Despite most reported symptoms in the literature related to FFP-mask wearing being nasal pressure injuries, mask-induced acne, pain in the ear lobe region and eczema [6,20], nasally specific symptoms should not be underestimated, due to their important impact on HC workers’ quality of life. The reported symptoms may be explained by the full adherence of the mask to the face and the consequent hot and wet environment that is generated [13]; the increased temperature could lead to vasodilation phenomena and as a consequence an increased blood supply to nasal mucosa, determining the hypertrophy of the nasal turbinate (nasal blockage) and submucosal inflammation that cannot be detected by nasal cytology. All these changes can alter ciliary motility, determining mucus stasis. This condition could be responsible for nasal symptoms, such as itching and rhinorrhea. However, in our study, as mentioned in Materials and Methods, ciliary motility was not studied to confirm this theory. Moreover, the referred discomfort could also be ascribed to the psychological impact of wearing such unwieldy PPE, rarely used before COVID pandemic [20].In fact, several studies demonstrate the psychological implication of wearing FFP2 masks; Carragher and Hancock observed that people are hard to recognize when wearing masks; emotional reading is substantially hampered, causing characteristic confusion of emotional states; and masks cause significant frequency-dependent transmission loss [21]; in a word, efficient communication is jeopardized [22]. All the psychological implications cited above could cause people to focus their attention on mild symptoms already experienced before the pandemic but not perceived as influencing their quality of life. Finally, it is important to remember how face masks also showed a strong impact on communication; in fact, a recent work highlights how among normal-hearing subjects, facial masks were more frequently held responsible for communication difficulties compared to social distancing, particularly due to an attenuation of sound volume and difficulties in rendering facial expressions [23]. The limits of this pilot study involve the small sample size and the short time of observation; indeed, the nasal epithelium could suffer from chronic remodeling due to a prolonged and continuous use of this personal protection equipment (PPE) and such modifications probably cannot be detected when focusing on such a short observation period only. Further examination of ciliary motility with a phase-contrast electronic microscope could provide more information about the pathogenesis of these symptoms.

## 5. Conclusions

The use of FFP2 is strongly recommended in order to protect from SARS-CoV-2 infection and this does not seem to be related to significant changes in nasal cytology over a short period of observation. However, further studies on a larger sample size and for a longer observational period should be performed in order to confirm these preliminary data.

## Figures and Tables

**Figure 1 healthcare-10-02365-f001:**
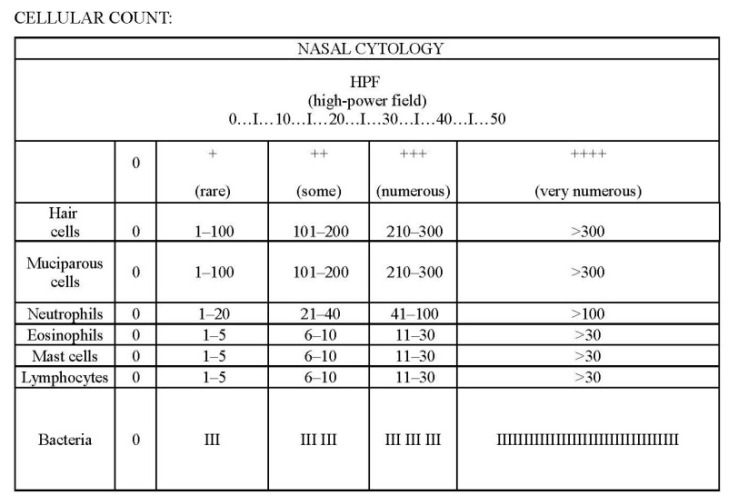
Rhinocytogram.

**Figure 2 healthcare-10-02365-f002:**
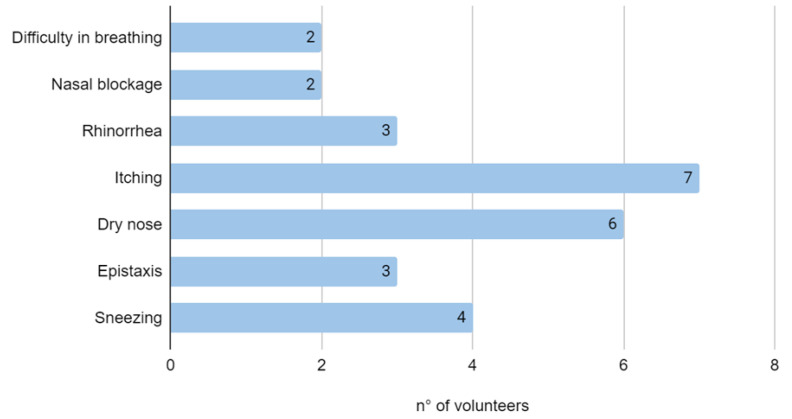
Symptoms reported by volunteers.

**Figure 3 healthcare-10-02365-f003:**
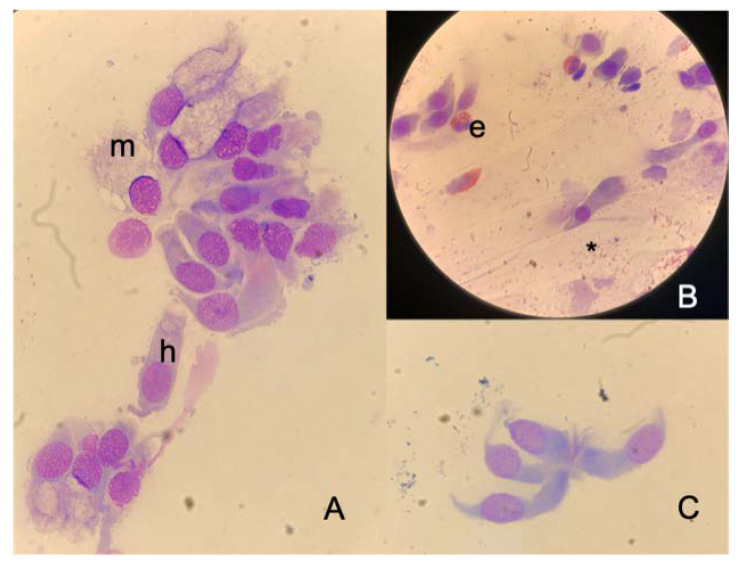
Nasal cytology: normal finding (**A**)with muciparous cells (m) and hair cells (h), inflammatory infiltration (**B**) by eosinophils (e) partially degranulated (*) and a group of hair cells (**C**)—May Grunwald Giemsa staining, original magnification ×1000.

**Table 1 healthcare-10-02365-t001:** Cytological features of volunteers at T0 and at T1.

Patient	Time of Observation	Eosinophils	Mast Cells	Neutrophils	Lymphocytes	Bacteria	Muciparous Cells	Hair Cells
1	T0	0	0	0	0	0	++	+++
	T1	0	0	0	0	0	++	+++
2	T0	0	0	0	0	0	++	++
	T1	0	0	0	0	0	++	++
3	T0	0	0	0	0	0	++	+++
	T1	0	0	0	0	0	++	+++
4	T0	+	0	+	0	0	+++	++
	T1	+	0	+	0	0	+++	++
5	T0	0	0	0	0	0	+++	++
	T1	0	0	0	0	0	+++	++
6	T0	0	0	0	0	0	+++	+++
	T1	0	0	0	0	0	+++	+++
7	T0	++	+	+	0	0	+	++
	T1	++	+	+	0	0	+	++
8	T0	0	0	+	0	0	++	+++
	T1	0	0	+	0	0	++	+++
9	T0	++	0	0	0	0	+	+
	T1	++	0	0	0	0	+	+
10	T0	0	0	0	0	0	++	++
	T1	0	0	0	0	0	++	++

+ rare; ++ some; +++ numerous. See Figure 1 for explanation.

## Data Availability

Not applicable.

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
