# Peer review of "Does Prolonged FFP2 Mask Use Cause Changes in Nasal Cytology? A Pilot Observational Study on Healthcare Workers"

_healthcare, 2022, doi:10.3390/healthcare10122365_

Round 1
Reviewer 1 Report
Do FFP2 mask prolonged use cause changes in nasal cytology? A pilot observational study on healthcare workers.
This article presents a pilot study on the implication of the prolonged use of FFP2 mask on nasal cytology.
Comments:
- In my opinion, this article should be accepted for publication after minor revision in “Healthcare”
- It is well developed, written in plain language, and easy to understand
- Results are clearly explained and documented in figures
- It is a pilot study so results might be improved with larger sample and longer time of “exposure” in order to obtain more consistent results. However, authors have largely declared their aim of initial observation and need of further studies.
- Face masks did not show only development of somatic symptoms but also psycho-social ones. A recent work highlights how among normal hearing subjects facial masks were more frequently held responsible for communication difficulties compared to social distancing, particularly due to an attenuation of sound volume and difficulties in rendering facial expressions. (reference: Giulia Elvira Malzanni, Chiara Canova, Rosa Alessia Battista, Paolo Malerba, Caterina Lerda, Sara Monica Angelone, Mario Bussi & Lucia Oriella Piccioni (2021) Restrictive measures during COVID-19 pandemic: the impact of face masks and social distancing on communication, physical and mental health of normal hearing subjects, Hearing, Balance and Communication, 19:3, 144-150, DOI: 10.1080/21695717.2021.1943788). This issue should be reported in the discussion.
Author Response
Thank you for the accurate revision of our article. We really appreciate the time you spent reading it and making suggestions in order to improve its quality. We took in consideration your suggestion of stressing the role of face masks in the emotional field in particular in communication difficulties between people. We add this aspect in discussion (lines 182-187).
Thank you very much
Kind regards

Reviewer 2 Report
Mask effect was studied only for a day time. It's not clear, for how much time the mask was used by the health workers. The cytology of the nose could have been changed since the beginning of pandemic. If there is no data about, The author's sentence; "Anyway, further studies on a larger sample size and for a longer observational period should be performed in order to confirm these preliminary data." is enough to explain drawbacks of study.
Author Response
Thank you for the accurate revision of our article. We really appreciate the time you spent reading it and making suggestions in order to improve its quality.
FFP2 mask were employed for at least 8 hours during the workshift. It is really probable that intrinsic changes in nasal mucosa occurred since the beginning of mask usage in 2020 with pandemic breakthrough. Unfortunately in literature there is no available data about this kind of change. Thi is why we concluded that further studies on a larger sample size and for a longer observational period should be performed.
Thank you very much
Kind regards

Reviewer 3 Report
The COVID19 pandemic has led to a sere of changes in daily work routines, including the use of face masks. The authors investigate the possible effects of prolonged use of FFP2 masks on the nasal mucosa of healthcare workers.
The manuscript is very well written, and it deals with a original and curious topic subject.
The study is well designed and appropriate.
Though, there are some things that need to be corrected.
The materials and methods are somewhat superficial. In the results you mention looking for nasal pathology and surgery, is this by any chance part of the study's exclusion criteria? Did you also include patients with allergic rhinitis or were they excluded?
Was the Rhinoprobe performed by different operators or by the same operator?
Were the slides analyzed by whom? Experienced pathologist, experienced researcher etc. Was the operator blinded for the study?
Did the "self-reported symptoms questionnaire" come from any article or was it part of an internal Hospital protocol?
Line 93: "Each cellular type is quantified and its numerosity 93 describes a severity score (e.g. eosinophils + : 1-5 cells, ++ : 6-10 cells, +++ : 11-30 cells)" I don't understand if it is an arbitrary score or jumps out of some alrticle; clarify and possibly cite.
The captions in Table 1, Figure 1 and Figure 2 are poor in detail. Also, the format of some captions is incorrect (they must go above the figure/table)
The text is clear and easy to read although minimal typos are present:
line 13 air way - airway
line 58 bilateral - bilaterally
line 75 suggest - suggests (if the subject is visualizating)
line 76 the observer will be.. - the observer can or could identify
line 92 can usedul - can be useful
line 173 several study - several studies
Same "the" are missing throughout the text , at the beginiing of sentences (e.g line 20: THE difficulty )
In the bibliography, the date of the articles should be put in boldface type
Author Response
Thank you for the accurate revision of our article. We really appreciate the time you spent reading it and making suggestions in order to improve its quality.
We are mortified about the lack of clarity in methods. The presence of allergic rhinitis and any other nasal pathology/surgery was considered as an exclusion criteria.
Samples were collected by the same operator in all 10 volouters and they were analyzed blinded by an otolaryngologist expert in nasal cytology (lines 63-64).
The self reported questionnaire is not a validated questionnaire, it is employed in daily practice during your examination in order to examine symptoms reported by patients.
The Line 93: "Each cellular type is quantified and its numerosity 93 describes a severity score (e.g. eosinophils + : 1-5 cells, ++ : 6-10 cells, +++ : 11-30 cells)" in extrapolated from Gelardi M, Iannuzzi L, Quaranta N, Landi M, Passalaqua G. NASAL cytology: practical aspects and clinical relevance.
Clinical and experimental allergy. 24 March 2016 DOI: 10,1111/cea,12730
We improved the captions you mentioned in the revision,corrected all spelling mistakes and bibliography inaccuracy.
Thank you very much
Kind regards

Reviewer 4 Report
The subject addressed in the manuscript entitled ”Do FFP2 mask prolonged use cause changes in nasal cytology? A pilot observational study on healthcare workers.” is very current and of great practical utility. The COVID-19 pandemic has led to important changes in the structure of personal protective equipment for healthcare workers (HC), imposing FPP2/N95 masks as a mandatory standard, especially when aerosol-generating procedures are involved, to reduce the risk of contagion. Prolonged wearing of FFP 2 masks by healthcare workers has resulted in nasal symptoms such as nasal obstruction, itching, nasal dryness, and so on.
This manuscript aims to determine whether prolonged use of FFP 2 masks in 8 hours work-shift among HC can cause changes in nasal cytology. The study was performed on a group of 10 HC (7 non-smokers and 3 smokers), without rhinosinusal pathology. Nasal samples were collected above the inferior turbinate, with a collection of mucus containing exfoliated cells from the nasal epithelium. All subjects studied, with one exception, experimented nasal symptoms after 8 hours of wearing the FFP2 mask. Comparative cytological data, between baseline and the end of the 8 hours work-shift showed no significant differences in nasal cytology, except for a slight decrease in the number of ciliated cells and muciparous cells, with no increase in the number of inflammatory cells.
The methodology is rigorous and thoroughly explained. Inclusion and exclusion criteria are detailed. The results are clearly formulated. The tables and figures accompanying the text are clearly explained and easy to understand.
The discussions are pertinent, looking for explanations for the nasal clinical symptomatology that appeared in the context of the prolonged wearing of the FFP 2 mask in HC. The authors point out that the increase in local temperature with consequent vasodilatation can cause hypertrophy of the nasal turbinates and thus explain the nasal obstruction. They discuss submucosal inflammation that cannot be detected by nasal cytology. This may be a limitation of the method used because by collecting the nasal mucus, only exfoliated cells can be studied, and a discrete inflammatory process in the submucosa may not be accompanied by an immediate change in the composition of the nasal mucus. The psychological consequences on HC because of long-term wearing of the FFP 2 mask are also invoked. Also, it might have been interesting to discuss whether the wearing of the FFP 2 mask can be interrupted at a predetermined time interval, for example at 2-3 hours, with a breath of fresh air during a work-shift for HC, a hypothesis that can be the premise of another prospective study.
References are appropriate and properly associated with the text. The English language is acceptable.
The limitations of this pilot study are described honestly. The results of the study represent a useful working premise for future studies conducted on larger groups and over a longer study period.
Author Response
Thank you for the accurate revision of our article. We really appreciate the time you spent reading it and making suggestions in order to improve its quality.
We will take in strong consideration your suggestion of interrupting FFP2 masks employment at predetermined intervals for further studies.
Thank you very much
Kind regards
